# Self-Supervised Learning of Representations for Space Generates Multi-Modular Grid Cells

**Rylan Schaeffer**[*]
Computer Science
Stanford University
rschaef@cs.stanford.edu

**Mikail Khona**[*]
Physics
MIT
mikail@mit.edu

**Tzuhsuan Ma**
Janelia Research Campus
Howard Hughes Medical Institute
mat@janelia.hhmi.org

**Cristóbal Eyzaguirre**
Computer Science
Stanford University
ceyzaguirre@stanford.edu

**Sanmi Koyejo**
Computer Science
Stanford University
sanmi@cs.stanford.edu

**Ila Rani Fiete**
Brain and Cognitive Sciences
MIT
fiete@mit.edu

## Abstract

To solve the spatial problems of mapping, localization and navigation, the mammalian lineage has developed striking spatial representations. One important spatial representation is the Nobel-prize winning grid cells: neurons that represent self-location, a local and aperiodic quantity, with seemingly bizarre non-local and spatially periodic activity patterns of a few discrete periods. Why has the mammalian lineage learnt this peculiar grid representation? Mathematical *analysis* suggests that this multi-periodic representation has excellent properties as an algebraic code with high capacity and intrinsic error-correction, but to date, *synthesis* of multi-modular grid cells in deep recurrent neural networks remains absent. In this work, we begin by identifying key insights from four families of approaches to answering the grid cell question: dynamical systems, coding theory, function optimization and supervised deep learning. We then leverage our insights to propose a new approach that elegantly combines the strengths of all four approaches. Our approach is a self-supervised learning (SSL) framework - including data, data augmentations, loss functions and a network architecture - motivated from a normative perspective, with no access to supervised position information. Without making assumptions about internal or readout representations, we show that multiple grid cell modules can emerge in networks trained on our SSL framework and that the networks generalize significantly beyond their training distribution. This work contains insights for neuroscientists interested in the origins of grid cells as well as machine learning researchers interested in novel SSL frameworks.

## 1 Introduction

Spatial information is fundamental to animal survival, particularly in animals that return to a home base. Mammalian brains exhibit a menagerie of specialized representations for remembering and

---

[*]Denotes equal contribution and co-first authorship.

37th Conference on Neural Information Processing Systems (NeurIPS 2023).

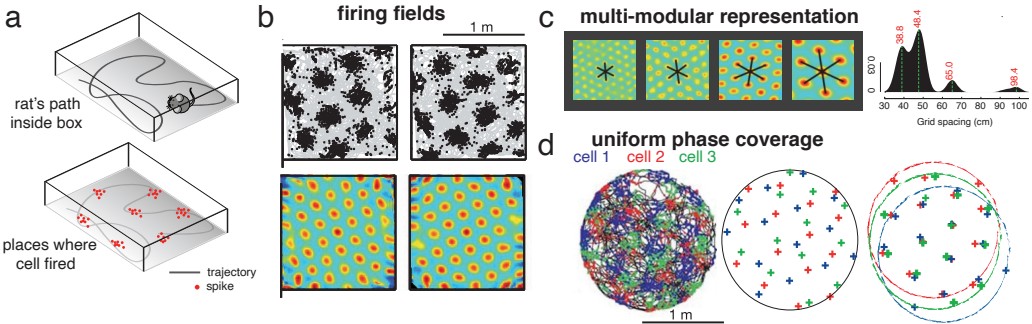

Figure 1: **Grid Cells in the Mammalian Medial Entorhinal Cortex.** (a) A grid cell fires at the vertices of a regular triangular lattice in space, regardless of direction or speed. (b) Examples of two grid cells' firing fields (top) and spatial autocorrelations (bottom). (c) The grid code is multimodular: across the population, grid cells cluster into discrete modules such that all cells within a module share the same spatial periodicity and orientation, and only a small number of spatial periods are expressed. (d) Grid cells within one grid module uniformly cover all phases. Figs b-d from [26, 59].

reasoning about space. Of these, *grid cells* (whose discovery earned a Nobel prize) are a striking and important example. Each grid cell fires in a spatially periodic pattern, at every vertex of a regular triangular lattice that tiles all two-dimensional environments [26] (Fig. 1ab) . This neural code for space appears bizarre since position is a local and aperiodic variable, but the neural representations are non-local and periodic. *Why has the mammalian brain learnt this peculiar representation?*

Broadly, there have been two directions to answering this question. The first direction uses theoretical analysis to deduce the mathematical properties that the grid code possesses [21, 56, 64, 40, 58]. This direction yields many insights about the coding properties of grid cells, but has not validated these insights by proposing an optimization problem under which grid cells emerge in deep recurrent neural networks. The second direction trains deep recurrent neural networks (RNNs) [16, 5, 54] with the goal of identifying the optimization problem that grid cells solve. However, recent work [48, 51] showed that these models require highly specific, hand-engineered and non-biological readout representations to produce grid cells; removing these representations results in loss of grid cells even though the optimization problem is equally well-solved.

In this paper, we make three contributions. First, we extract critical insights from prior models of the grid code spanning diverse fields including coding theory, dynamical systems, function optimization and supervised deep learning. Second, we use those insights to propose a new self-supervised learning (SSL) framework for spatial representations, including data, data augmentations, loss functions and network architecture. Third, we show that multi-modular grid cells can emerge as optimal solutions to our SSL framework, with no hand-engineered inputs, internals or outputs. We also perform ablations to show under what constraints grid cells do and do not emerge.

By creating a minimal, theoretically-motivated and developmentally plausible learning problem for path-integrating deep recurrent networks through SSL, we close the loop from the discovery of biological grid cells, to the study of their theoretical properties, to their artificial genesis.

## 2    Background

**Grid cells in the mammalian medial entorhinal cortex**    Grid cells [26] are neurons found in the mammalian medial entorhinal cortex tuned to represent the animal's spatial location as it traverses 2D space. Each grid cell fires if and only if the animal is spatially located at any vertex of a regular triangular lattice that tiles the explored space, regardless of the speed and direction of movement through space (Fig. 1ab). As a population, grid cells exhibit several striking properties that provide support for a specialized and modular circuit: Grid cells form discrete modules (clusters) such that all cells within a module share the same spatial periodicity and orientation, while different modules express a small number of discretely different spatial periods (Fig. 1c) [59]. Grid cells in the same module (i.e., sharing the same lattice periodicity) exist for all phase shifts [68] (Fig. 1d).

**Self-Supervised Learning (SSL) and its applications in Neuroscience**    SSL is a subset of unsupervised learning in which data is used to construct its own learning target [2, 13, 27, 61, 10, 11, 14, 69, 19]. Broadly, SSL methods belong to one of three families: deep metric learning, self-distillation or canonical correlations analysis; we refer the reader to a recent review [4]. SSL is increasingly gaining popularity as a normative approach in neuroscience since SSL mitigates the need for large-scaled supervised data that biological agents do not possess. SSL has been used in vision to explain the neural representations of ventral visual stream [32, 70], dorsal visual stream [41] and an explanation for the emergence of both [3]. In this paper, we provide the first model of SSL applied to the mammalian brain's spatial representations with an emphasis on sequential data and recurrent networks that to the best of our knowledge is not well explored in either the machine learning or neuroscience literature.

## 3   Notation and Terminology

We consider temporal sequences of three variables: spatial positions $x_t \in \mathbb{R}^2$, spatial velocities $v_t \in \mathbb{R}^2$ and neural representations on the surface of the positive orthant of the $N$-dimensional sphere $g_t \in \mathbb{S}_{\geq 0}^{N-1} \overset{\text{def}}{=} \mathbb{G}$. Note that any neural representation $g \in \mathbb{G}$ has unit norm $||g||_2 = 1$. We interchangeably refer to representations as states, codes or embeddings. We denote sequences of any of these variables from time $t = 1$ to $T$ as $(\cdot_1, \cdot_2, ..., \cdot_T)$, e.g., $(v_1, v_2, ..., v_T)$.

## 4   Insights from Grid Cell Modeling Approaches

The question of why the mammalian brain uses a high-dimensional, multiperiodic code to encode a low-dimensional, aperiodic variable has led researchers to theoretically analyze the grid code's properties from multiple mathematical perspectives: coding theory, dynamical systems, function optimization, and supervised deep learning. In this section, we extract key insights that suggest a convergence towards a SSL approach.

### 4.1   Insights from Coding Theory

Many interrelated coding-theoretic properties of grid cells have been pointed out before: **Exponential capacity**: The coding states form a space-filling curve (1D) (Fig. 2a) or space-filling manifold (2D) with well-separated coding states in the full coding volume defined by the number of cells. Linearly many distinct grid modules and therefore linearly many neurons generate exponentially many unique coding states for generic choices of periods [21, 56]. The grid code enables high dynamic-range representation (i.e., large value of range divided by resolution) of space using individual parameters that do not vary over large ranges [21] (Fig. 2d-e). **Separation**: Excluding the closest spatial locations, the coding manifold for spatial positions is interleaved in the coding volume, such that relatively nearby states are mapped to faraway locations (Fig. 2b) and relatively nearby spatial locations are mapped to faraway states (Fig. 2c); this gives the grid code strong error-correcting properties [56]. **Decorrelation**: Grid cells maximally decorrelate their representations of space beyond a small scale that is roughly half the period of the smallest module, so that all correlations are roughly equally small beyond that scale [21, 56] (Fig. 2f). **Equinorm coding states**: The grid coding states are equi-norm and lie on the surface of a hypersphere. **Ease of update**: The grid code permits carry-free updating arithmetic so that individual modules update their states without communicating with each other, meaning updates can be computed in parallel rather than sequentially [21]. **Algebraic**: The grid code is a faithful algebraic code [21, 56], in that the grid code for composition of displacements is equal to the composition of grid codes for each of the displacements. This property enables the grid code to self-consistently represent Euclidean displacements (Fig. 2g). **Whitened representation**: All modules (coding registers or phases) update by similar amounts for a given displacement in real space, as a consequence of module periods being similarly sized (in contrast with fixed-base number system representations, where the "periods" for each register are hierarchically organized in big steps as powers of the base, and the representation is not whitened).

These properties are all intrinsic to the grid code, without regard to other types of spatial neurons, e.g. place cells. The lesson is that the grid code likely originates from coding-theoretic properties, not from predicting place cells (c.f., [5, 55]). The challenge before us is identifying which subset of these partially inter-related mathematical coding-theoretic properties of the grid code - all useful for the function of grid cells - forms the minimally sufficient subset for grid cell emergence.

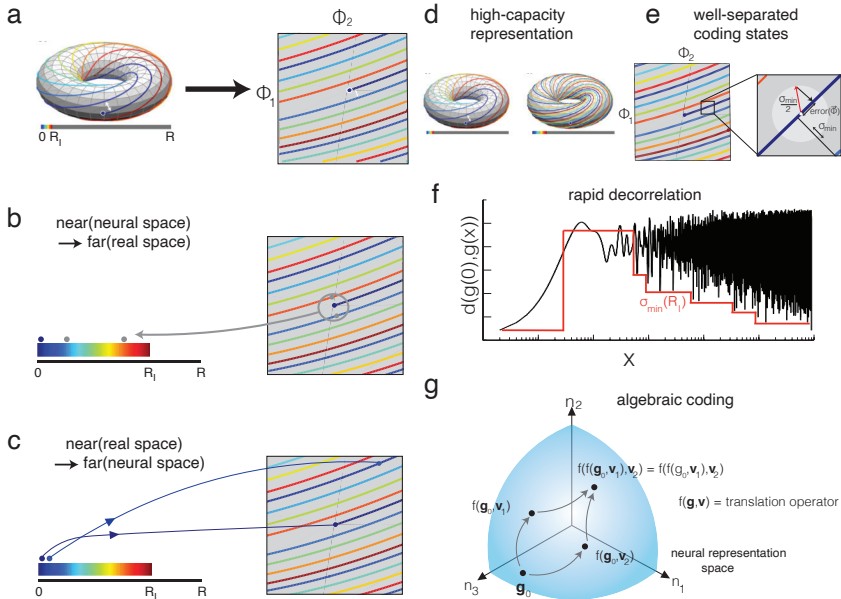

Figure 2: **Mammalian grid cells and their coding theoretic properties:** Schematics in (a)-(e) depict the coding of 1D rather than 2D space for simplicity, with phases of two grid modules. (a) Continuous locations translate to a continuous coding line that wraps around the torus defined by the two grid module phases. Right: torus cut open to a sheet with boundary conditions. (b) Near-to-far mapping from phases to locations. (c) Near-to-far mapping from locations to phases. (d-e) High capacity and separability: For generic choices of grid module periods, the code is a space-filling curve on the phase torus (d), with a capacity (before curves overlap for finite-thickness curves) that grows exponentially with the number of modules. The coding lines tend to be well-separated for any range of real-space being encoded (2 shown) because of the interleaving property of the code (e). (f) The distance between coding representations $g(0)$ and $g(x)$ very quickly increases (around $\lambda/2$ where $\lambda$ is the scale of the smallest module.) (g) Path-independent encoding of displacements.

## 4.2 Insights from Dynamical Systems

Consider the following general framework for defining a neural code that is dynamically updated on the basis of a velocity signal: An agent starts with some neural representation $g_0 \in \mathbb{G}$, then moves along a spatial trajectory given by the sequence of velocities $(v_1, ..., v_T)$. The sequence of neural representations $(g_1, ..., g_T)$ is updated based on velocity through the (arbitrary, non-linear and possibly time-dependent) dynamical translational operation $f : \mathbb{G} \times \mathbb{R}^2 \to \mathbb{G}$: $g_t \stackrel{\text{def}}{=} f(g_{t-1}, v_t)$. In order for the neural representations to be good codes for Euclidean space, what properties should $f$ and $\mathbb{G}$ possess? The first property is that the representations should be **path invariant**: $g_T$ should be the same regardless of whether the agent took path $v_1, ..., v_T$ or path $v_{\pi(1)}, ..., v_{\pi(T)}$ for any permutation $\pi : [T] \to [T]$. Specifically, we desire that $\forall g, v_i, v_j, \quad f(f(g, v_i), v_j) = f(f(g, v_j), v_i)$. Note that path independence automatically implies that the coding states form a **continuous attractor** [21], because when the velocity $v = 0$, the state should remain constant: $\forall g, \quad f(g, v = 0) = g$. Further, the grid cell representation is translationally invariant [39, 31]: representations at different positions are related by a shift operator that only depends on the relative separation between the positions, i.e., the correlational structure $\langle g(x)g(x+v) \rangle_x = f(v)$, where $f$ is a function only of $v$. The lessons are thus twofold: the data and loss function(s) should together train the network to learn path invariant representations and continuous attractor dynamics, and the dynamical translation operation of the recurrent neural networks should architecturally depend on the velocity.

## 4.3 Insights from Spatial Basis Function Optimization

A third approach to explaining grid cells is via optimization of spatial basis functions. The spatial basis function optimization approach is a *non-dynamical* one that posits a *spatial* neural code $g(x)$ of

a particular parametric form, e.g., $\boldsymbol{g}(\boldsymbol{x}) \stackrel{\text{def}}{=} \boldsymbol{a}_0 + \sum_d \boldsymbol{a}_d \sin(\boldsymbol{k}_d \cdot \boldsymbol{x}) + \sum_d \boldsymbol{b}_d \cos(\boldsymbol{k}_d \cdot \boldsymbol{x})$. One can then define a loss function and optimize parameters to discover which loss(es) results in grid-like tuning, possibly under additional constraints [39, 18]. This functional form is used because it reflects the constraints of path-integration: if one path-integrates with action-dependent weight matrices, then one *must* use that functional form [18]. While sensible, these insights have not yet crossed to deep recurrent neural networks. The challenges are that recurrent networks given velocity inputs are not explicit functions of space, and their representations need not be comprised of Fourier basis functions. We seek to bring such insights, which requires discovering how to train recurrent networks that simultaneously learn a consistent representation of space (i.e., learn a continuous attractor), while at the same time, shape the learnt representations to have desirable coding properties.

## 4.4 Insights from Supervised Deep Recurrent Neural Networks

Deep recurrent neural networks sometimes learn grid-like representations when trained in a supervised manner on spatial target functions [5, 54, 23, 65, 22, 67], and studies regarding the successor representation in reinforcement learning [57, 42] show similar findings. These results share a common origin based on the eigendecomposition of the spatial supervised targets (putatively: place cells) which results in periodic vectors and was earlier noted in shallow nonlinear autoencoders [17]. This understanding was used by [54] to hand-design better supervised targets.

While high profile, this line of work has fundamental shortcomings. Firstly, because the networks learn to regurgitate whatever the researcher encodes in the spatial target functions, the learned representations are set by the researcher, not by the task or fundamental constraints [49]. Secondly, by assuming the agent already has access to its spatial position (via the supervised targets), this approach bypasses the central question of how agents learn how to track their own spatial position. Thirdly, previous supervised path-integrating recurrent neural networks either do not learn multiple grid modules or have them inserted via implementation choices, oftentimes due to statistics in the supervised targets [54, 49]. Fourthly, supervised networks fare poorly once the spatial trajectories extend outside the training arena box (i.e., where supervised targets have not been defined).

The lesson we extract from this is to *remove supervised targets altogether, and instead directly optimize the neural representations via self-supervised learning.* Supervised networks pick up whatever statistics exist in the supervised targets, and struggle to generalize beyond their supervised targets, meaning that if we wish to avoid putting grid cells in by hand or if we wish to obtain networks that generalize, we must remove the supervised targets altogether. Previous work lightly explored this direction by introducing unsupervised loss terms, but kept the supervised targets [22, 67]. In this work, we fully commit to this lesson and take a purely SSL approach.

# 5 Self-Supervised Learning of Representations for Space

We deduce that learning useful representations of space is a SSL problem requiring an appropriate SSL framework (comprising training data, data augmentations, loss functions and architecture). We hypothesize that three loss functions, extracted from the coding, path integration and optimization perspectives synthesized above, when combined with particular data and recurrent network architecture, will generate multiple modules of grid cells. See App. A for experimental details.

## 5.1 Data Distribution and Data Augmentations

For each gradient step, we sample a sequence of $T$ velocities $(\boldsymbol{v}_1, \boldsymbol{v}_2, ..., \boldsymbol{v}_T)$, with $\boldsymbol{v}_t \sim_{i.i.d.} p(\boldsymbol{v})$, then construct a batch by applying $B$ randomly sampled permutations $\{\pi_b\}_{b=1}^B$, $\pi_b : [T] \to [T]$ to the sequence of velocities to obtain $B$ permuted velocity trajectories; doing so ensures many intersections between the trajectories exist in each batch (SI Fig. 8b). We feed each trajectory through a recurrent network (Sec. 5.2) with shared initial state $\boldsymbol{g}_0$, producing a sequence of neural representations $(\boldsymbol{g}_{\pi_b(1)}, \boldsymbol{g}_{\pi_b(2)}, ..., \boldsymbol{g}_{\pi_b(T)})$. This yields a batch of training data for a single gradient step.

$$\mathcal{D}_{\text{gradient step}} = \left\{ \left( \boldsymbol{v}_{\pi_b(1)}, \boldsymbol{v}_{\pi_b(2)}, ..., \boldsymbol{v}_{\pi_b(T)} \right), \left( \boldsymbol{g}_{\pi_b(1)}, \boldsymbol{g}_{\pi_b(2)}, ..., \boldsymbol{g}_{\pi_b(T)} \right) \right\}_{b=1}^B \tag{1}$$

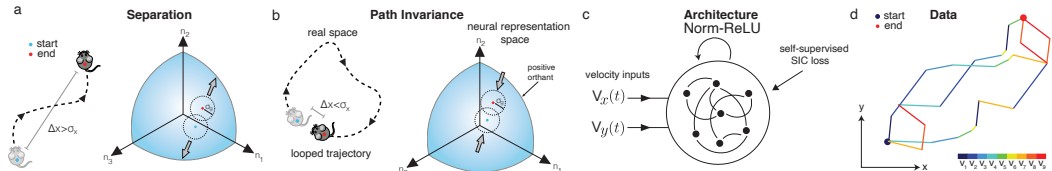

Figure 3: **A self-supervised framework for learning representations of space.** (a) Separation incentivizes neural representations corresponding to different spatial locations to be well-separated. (b) (Path) Invariance incentivizes neural representations corresponding to the same spatial location to match. (c) The architecture of the recurrent neural network used. Inputs are 2D Cartesian velocities $\boldsymbol{v}_t \in \mathbb{R}^2$ and the non-linearity is Norm-ReLU. No positional readout exists. (d) For each batch, input trajectory velocities are drawn i.i.d. from a uniform distribution, then randomly permuted to create a batch; three trajectories shown.

## 5.2 Recurrent Deep Neural Network Architecture

Motivated by our insights from dynamical systems (Sec. 4.2), we use an RNN (SI Fig. 8a) with interaction matrix given by:

$$W(\boldsymbol{v}_t) \stackrel{\text{def}}{=} MLP(\boldsymbol{v}_t) \tag{2}$$

where $\boldsymbol{v}_t \in \mathbb{R}^2$ is the velocity input to the recurrent network and $W : \mathbb{R}^2 \to \mathbb{R}^{N \times N}$ for $N$ hidden RNN units. The dynamics of the recurrent network are defined as:

$$\boldsymbol{g}_t \stackrel{\text{def}}{=} \sigma(W(\boldsymbol{v}_t)\,\boldsymbol{g}_{t-1}) \tag{3}$$

The neural representations $\boldsymbol{g}_t$ are enforced to be non-negative and unit-norm via the non-linearity $\sigma(\cdot) \stackrel{\text{def}}{=} Norm(ReLU(\cdot)) = ReLU(\cdot)\,/\,||ReLU(\cdot)||$. Constraining representations to be constant norm is a common technique in SSL to avoid representation collapse [4] and also was used in previous grid cell modelling work [67]. By choosing this particular architecture, we ensure that representations at different points in space are related only by a function of velocity, thereby ensuring that the transformation applied to the state is the same so long as the velocity is the same.

## 5.3 Loss Functions

**Separation Loss** Neural representations corresponding to different spatial locations, regardless of trajectory, should be well-separated (Fig. 3a). Here, "different spatial locations" is with respect to spatial length scale $\sigma_x > 0$ and "well-separated" is with respect to neural length scale $\sigma_g > 0$.

$$\mathcal{L}_{Sep} \stackrel{\text{def}}{=} \sum_{\substack{\forall \pi_b, \pi_{b'}, t, t': \\ ||\boldsymbol{x}_{\pi_{b'}(t)} - \boldsymbol{x}_{\pi_b(t')}||_2 > \sigma_x}} \exp\left(-\frac{||\boldsymbol{g}_{\pi_{b'}(t)} - \boldsymbol{g}_{\pi_b(t')}||_2^2}{2\sigma_g^2}\right) \tag{4}$$

**Path Invariance Loss** Neural representations corresponding to the same spatial location, regardless of trajectory, should be identical (Fig 3b), i.e., neural representations should be *invariant* to the particular path the agent takes. If two trajectories are permuted versions of one another such that they begin at the same position and end at the same position (measured with respect to a given spatial length scale $\sigma_x > 0$), the neural representations after both trajectories should be attracted together.

$$\mathcal{L}_{Inv} \stackrel{\text{def}}{=} \sum_{\substack{\forall \pi_b, \pi_{b'}, t, t': \\ ||\boldsymbol{x}_{\pi_{b'}(t)} - \boldsymbol{x}_{\pi_b(t')}||_2 < \sigma_x}} \left|\left|\boldsymbol{g}_{\pi_b(t)} - \boldsymbol{g}_{\pi_{b'}(t')}\right|\right|_2^2 \tag{5}$$

**Capacity Loss** Neural representations should be as high capacity as possible. That means that given a finite neural representation space, the number of unique spatial positions that the neural representations can uniquely encode should be maximized. In our particular setting, because our representations lie on the hypersphere, we can incentivize high capacity in the following manner:

$$\mathcal{L}_{Cap} \stackrel{\text{def}}{=} -\left|\left|\frac{1}{B\,T}\sum_{\pi_b, t}\boldsymbol{g}_{\pi_b(t)}\right|\right|_2^2 \tag{6}$$

a

1m

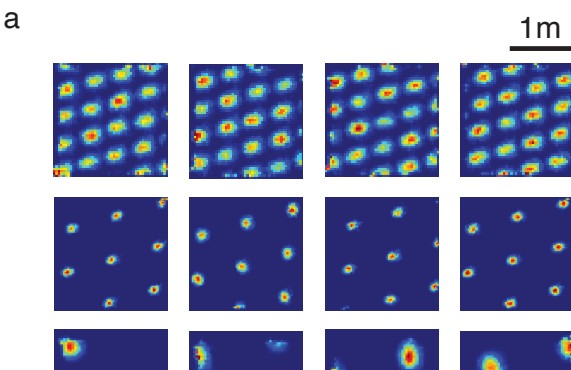

Figure 4: **Multiple modules of grid cells in trained RNN:** (a) The RNN's hidden units learn three discrete modules of spatially periodic representations. Four cells from each module are displayed.

This capacity loss pushes together all neural embeddings, regardless of path traveled or point in time, encouraging as little of the embedding space to be allocated for the training data as possible.

**Conformal Isometry Loss**   Consider a velocity $v_t$ that shifts the neural code from $g_t$ to $g_{t+1}$. A neural code is conformally isometric if the distance between the neural codes remain constant if $v_t$ changes direction (but not magnitude). Motivated by [22, 67], we propose a novel and simpler conformal isometry loss that minimizes the variance in ratios of changes in code space to changes in physical space for small velocities:

$$\mathcal{L}_{ConIso} \overset{\text{def}}{=} \mathbb{V}\Big[\Big\{\frac{||\boldsymbol{g}_t - \boldsymbol{g}_{t-1}||}{||\boldsymbol{v}_t||}\Big\}_{t\,:\,0<||\boldsymbol{v}_t||<\sigma_x}\Big], \tag{7}$$

where $\mathbb{V}$ denotes the variance. Intuitively, $\mathcal{L}_{ConIso}$ asks that neural representations change by some constant amount for the same magnitude of velocity, regardless of the velocity's direction. Our overall self-supervised loss is then a weighted combination of these four losses:

$$\mathcal{L} \overset{\text{def}}{=} \lambda_{Sep}\,\mathcal{L}_{Sep} \; + \; \lambda_{Inv}\,\mathcal{L}_{Inv} \; + \; \lambda_{Cap}\,\mathcal{L}_{Cap} + \; \lambda_{ConIso}\mathcal{L}_{ConIso} \tag{8}$$

We term this framework Separation-Invariance-Capacity (**SIC**). This SIC framework is naturally suited to the setting where neural representations are generated by a deep recurrent network which receives velocity inputs. Unlike previous (supervised) approaches, our SIC framework requires no information about *absolute* spatial position, only *relative* spatial separation, and does not require any tuned supervised target, thus mitigating the shortcomings of previous works identified recently [48].

## 6   Experimental Results

### 6.1   Emergence of multiple modules of periodic cells

Under our SIC SSL framework, we find that trained networks learn neurons with spatially multi-periodic responses. In each run, most neurons in the network exhibit periodic tuning. Moreover, the network clearly exhibits a few discrete periods. We used grid search over hyperparameters with roughly $1/2$ an order of magnitude variation for key hyperparameters ($\sigma_x, \sigma_g, \lambda_{Sep}, \lambda_{Inv}, \lambda_{Cap}$) and found that multiple modules robustly emerge across this range. Computational limitations prevented us from sweeping more broadly. Example units from a run are shown in Fig. 4. Below, we characterize different facets of the results.

### 6.2   Generalization to input trajectory statistics

A drawback in previous (supervised) deep learning grid cell works was poor generalization when the networks are evaluated in arenas larger than their training arenas. Grid cells in our networks

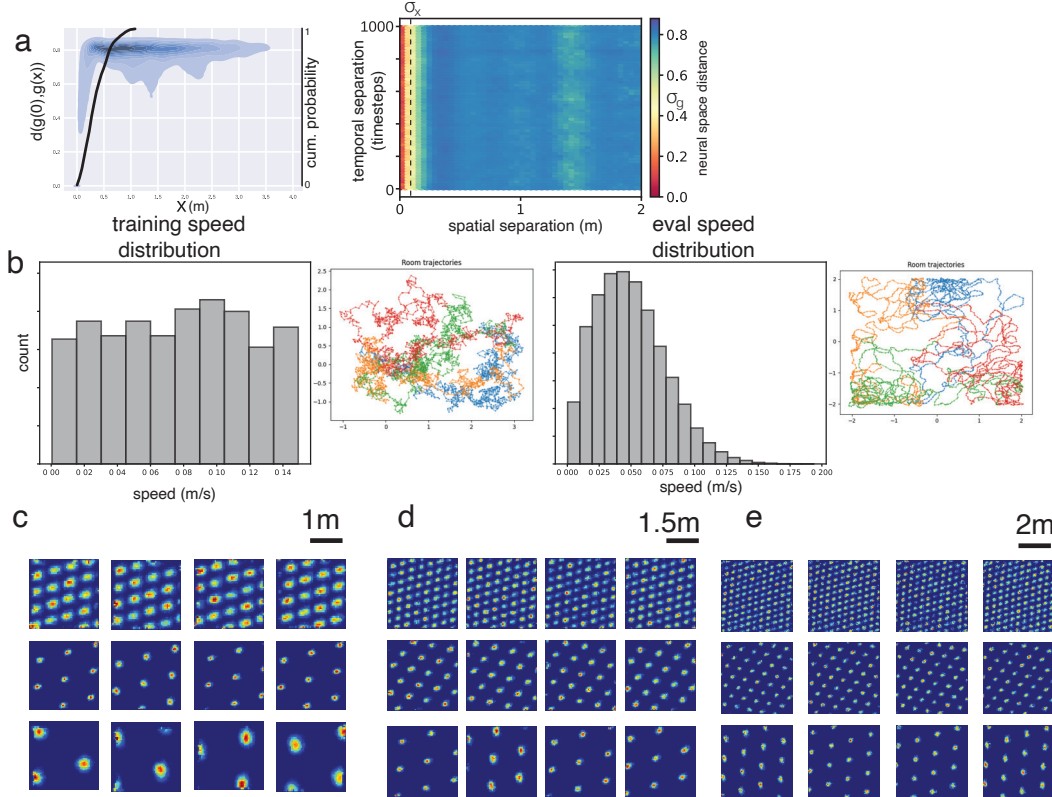

Figure 5: **Grid tuning generalizes to different velocity statistics and to environments much larger than the training environment.** (a) [left] A kernel density estimate plot of pairwise distances in neural space for randomly chosen pairs of points in real space showing rapid decorrelation over a length scale that is $\approx \sigma_x$. A cumulative CDF of the pairwise distance distribution over 1 training batch (black), showing that the network generalizes to environment sizes much larger than it has seen during training. [right] A plot of pairwise neural space distance as a function of both spatial separation and temporal separation. Red vertical line at 0 spatial separation denotes perfect path integration. (b) Training trajectory statistics (left) and evaluation trajectory statistics (right) (c-e) Ratemaps for trajectories in boxes of sizes 2m, 3m, 4m.

generalize to arenas many times larger than the distances explored during training (Fig. 5c-e) (training trajectories extend $<1.2$m from the starting position (Fig. 5a)). Secondly, the training trajectories were drawn i.i.d. from a uniform velocity distribution, Fig. 5b (left), while evaluation trajectories were constructed to be smoother and avoid the walls of a confined arena (Fig. 5b, left). As found in experiments, the periodicities of the emergent grid cells remained the same, regardless of the size of the arena (Fig 5c-e).

## 6.3 State space analysis of trained RNNs

We turn to a more-detailed characterization of the grid cells in each module and their cell-cell relationships/population states. For a big enough set of cells for comparison with experimental results, we needed sufficiently many co-modular cells and so used the network corresponding to Fig. 7a, in which most of the units formed grids of the same period.

We found that cells with similar periods were tightly clustered to have essentially identical periods and orientations (Fig. 6a). Moreover, this population tiled the space of all phases (the Fourier PSD contains a hexagonal structure with wavevectors $k_1, k_2, k_3$; we defined three non-orthogonal phases satisfying $\sum_{a=1,2,3} \phi_i^a = 0 \mod 2\pi$ for each unit from its ratemap $r(\boldsymbol{x})$: $\phi_i^a = \text{phase}\left[\int d\boldsymbol{x} e^{i\boldsymbol{k}_a \cdot \boldsymbol{x}} r_i(\boldsymbol{x})\right]$), exhibiting a uniform phase distribution (Fig. 6b).

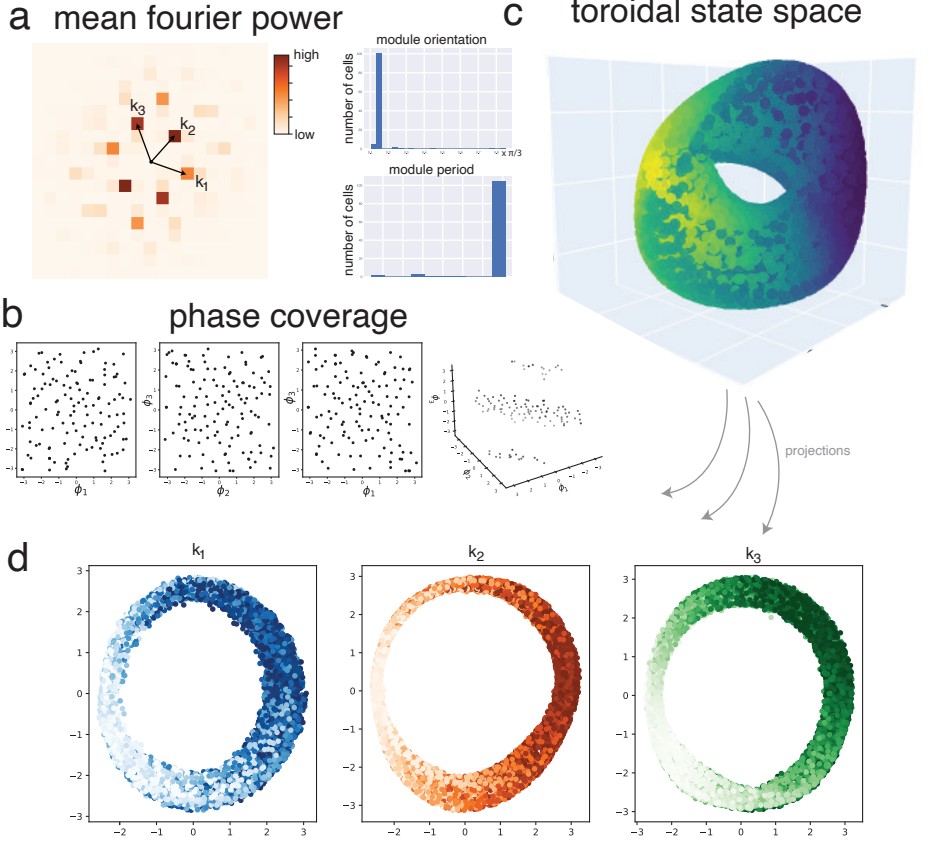

Figure 6: **Analysis of low-dimensional structure in one of the grid cell modules:** (a) (left) Mean Fourier power spectral density of all 128 units in the module, showing hexagonally arranged peaks. (right) Histogram of cell periods and orientations. (b) Distribution of phases with phases $\phi_1, \phi_2, \phi_3$ calculated by evaluating the Fourier transform of each ratemap along each lattice vector. (c) Cells from one module exhibit a toroidal manifold (visualization by non-linear dimensionality reduction (App D)). (d) Projection of the state space manifold onto the three principal directions defined by vectors $k_1, k_2, k_3$ shows 3 rings.

In theoretical models and real data, grid cells exhibit low-dimensional dynamics and the states of all cells in a module lie on a 2D torus. We used spectral embedding [47] to find that the emergent grid cell population states also fell on a toroidal manifold (Fig. 6a). Following [55], we projected the activity manifold onto 3 axes defined by each set of phases calculated for each unit. This revealed the existence of 3 rings, Fig. 6d, confirming that the attractor manifold has a 2d twisted torus topology.

**Ablation experiments**  We also ablated each of the three loss terms, the data augmentation, i.e., trajectory permutations and the hyperparameter $\sigma_g$ to reveal when multiple modules of grid cells changes to a single module and then to a place-cell like code (Fig 7a-f).

## 7   Discussion & Future Directions

In this work, we extracted insights from previous first principles coding theoretic properties of grid cells and mechanistic models that produce grid-like firing. We then proposed a novel set of self-supervised loss functions that with the architectural constraint of a recurrent neural network can produce multiple modules of grid cells when trained on a spatial navigation task. In doing so, we have highlighted the distinction between separation and capacity for a neural representation and shown that optimizing for capacity leads to efficient representations that are better for generalization to larger amounts of test data.

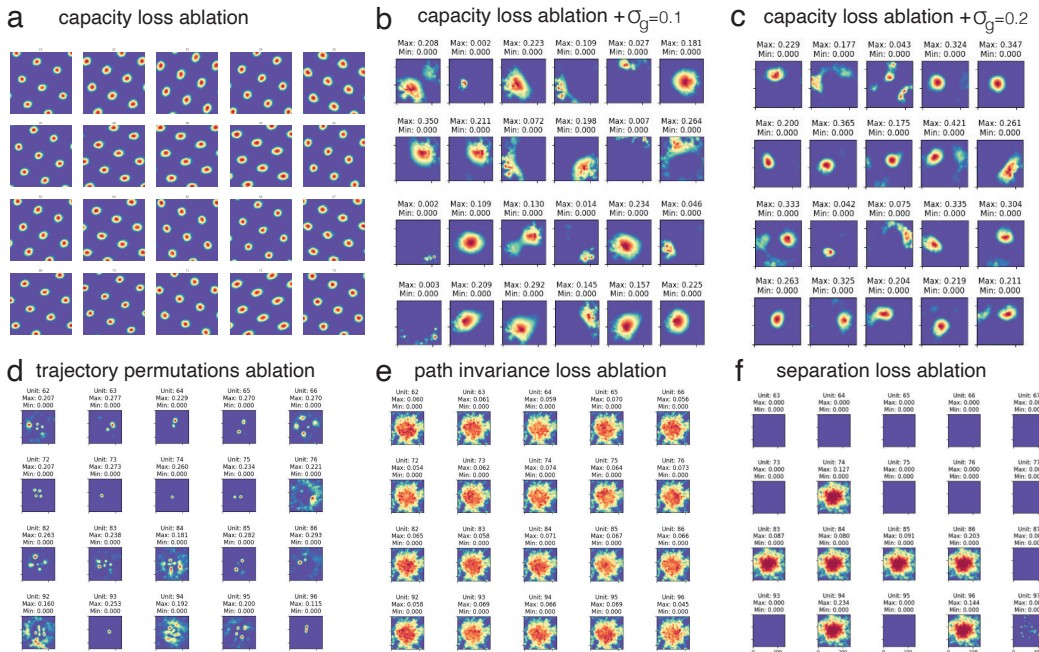

Figure 7: **Ablation experiments:** (a) Removal of capacity loss leads to a single module of grid cells. (b-c) Reducing $\sigma_g$ makes the representation place cell-like, with size of place fields decreasing with increasing $\sigma_g$. (d-f) Ablating other key ingredients leads to loss of spatial tuning.

Looking to the future, we see four broad directions, two in neuroscience and two in machine learning. Firstly, we believe significantly more work is necessary to understand these trained recurrent networks and connect them back to grid cells. Key questions include: What components of the data, augmentations, losses and architecture control the number of modules, the distribution of the module periods, the period ratios, the structure (square or hexagonal) of the lattices? What do the state spaces of these networks look like? Can we reverse-engineer the learnt mechanisms, using mechanistic analyses [50, 30, 37]? Another interesting question is what underlying connectivity leads to discrete modular responses [29]. How do the learnt representations change when exposed to experimental manipulations, e.g., environmental deformations or rewards [33, 34, 35, 28, 8, 9, 7]? What new experiments can these networks suggest in biological agents? Secondly, we see many opportunities in neuroscience broadly. The biological brain cares about high capacity representations, and grid cells have been implicated in many diverse cognitive and non-spatial tasks [15, 1, 6, 45, 63, 44]. How might our grid cell recurrent networks, and more generally, SSL principles be applied to drive computational neuroscience forward?

Thirdly, we are excited to push our SIC framework further towards machine learning. We envision empirically exploring whether our SIC framework can improve performance in other popular SSL domains such as vision and audition, as well as mathematically characterizing the relationship between SIC and other SSL frameworks. Fourthly, one puzzling aspect of our capacity loss is that it can be viewed as *minimizing* the entropy of the neural representations in representation space, which flies against common information theoretic intuition arguing for *maximizing* the entropy of the representations [66, 43, 62, 36, 53, 52]. In some sense, the capacity loss is akin to a "dual" minimum description length problem [25]: instead of maximizing compression by having a fixed amount of unique data and trying to minimize the code words' length, here we maximize compression by having fixed length code words and trying to maximize the amount of unique data. How can this apparent contradiction be resolved?

## 8 Acknowledgements

IRF is supported by the Simons Foundation through the Simons Collaboration on the Global Brain, the ONR, the Howard Hughes, Medical Institute through the Faculty Scholars Program and the K. Lisa Yang ICoN Center. SK acknowledges support from NSF grants No. 2046795, 1934986, 2205329, and NIH 1R01MH116226-01A, NIFA 2020-67021-32799, the Alfred P. Sloan Foundation, and Google Inc. RS is supported by a Stanford Data Science Scholarship. MK is supported by a MathWorks Science Fellowship and the MIT Department of Physics.

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

# A  Experimental Details

**Architecture and training data + augmentations**

Our code was implemented in PyTorch [46] and PyTorch Lightning [20]. Hyperparameters for our experiments are listed in Table 1. Our code will be made publicly available upon publication.

| Hyperparameters | Values |
|---|---|
| Batch size | 130 |
| Trajectory length | 60 |
| Velocity sampling distribution | $v_t \in \mathbb{R}^2 \sim_{i.i.d.} \text{Uniform}^2(-0.15, 0.15)$ meters |
| RNN nonlinearity | $Norm(ReLU(\cdot))$ |
| Number of RNN units | 128 |
| Number of MLP layers | 3 |
| Spatial length scale $\sigma_x$ | 0.05 meters |
| Neural length scale $\sigma_g$ | 0.4 |
| Separation loss coefficient $\lambda_{Sep}$ | 1.0 |
| Invariance loss coefficient $\lambda_{Inv}$ | 0.1 |
| Capacity loss coefficient $\lambda_{Cap}$ | 0.5 |
| Optimizer | AdamW [38] |
| Optimizer scheduler | Reduce Learning Rate on Plateau |
| Learning rate | 2e-5 |
| Gradient clip value | 0.1 |
| Weight decay | None |
| Accumulate gradient batches | 2 |
| Number of gradient descent steps | 2e6 |

Table 1: Hyperparameters used for training the networks.

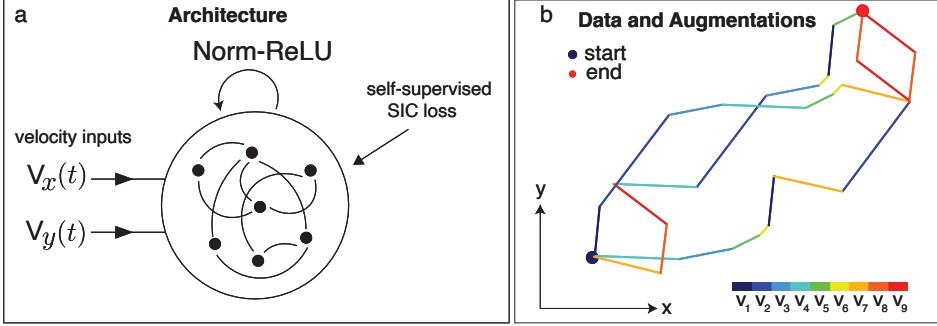

Figure 8: **A trained recurrent neural network learns multiple modules of grid cells.** (a) The architecture of the recurrent neural network used. Inputs are 2D Cartesian velocities $v_t \in \mathbb{R}^2$ and the non-linearity is Norm-ReLU. No positional readout exists. (b) For each batch, input trajectory velocities are drawn i.i.d. from a uniform distribution, then randomly permuted to create a batch; 3 trajectories shown.

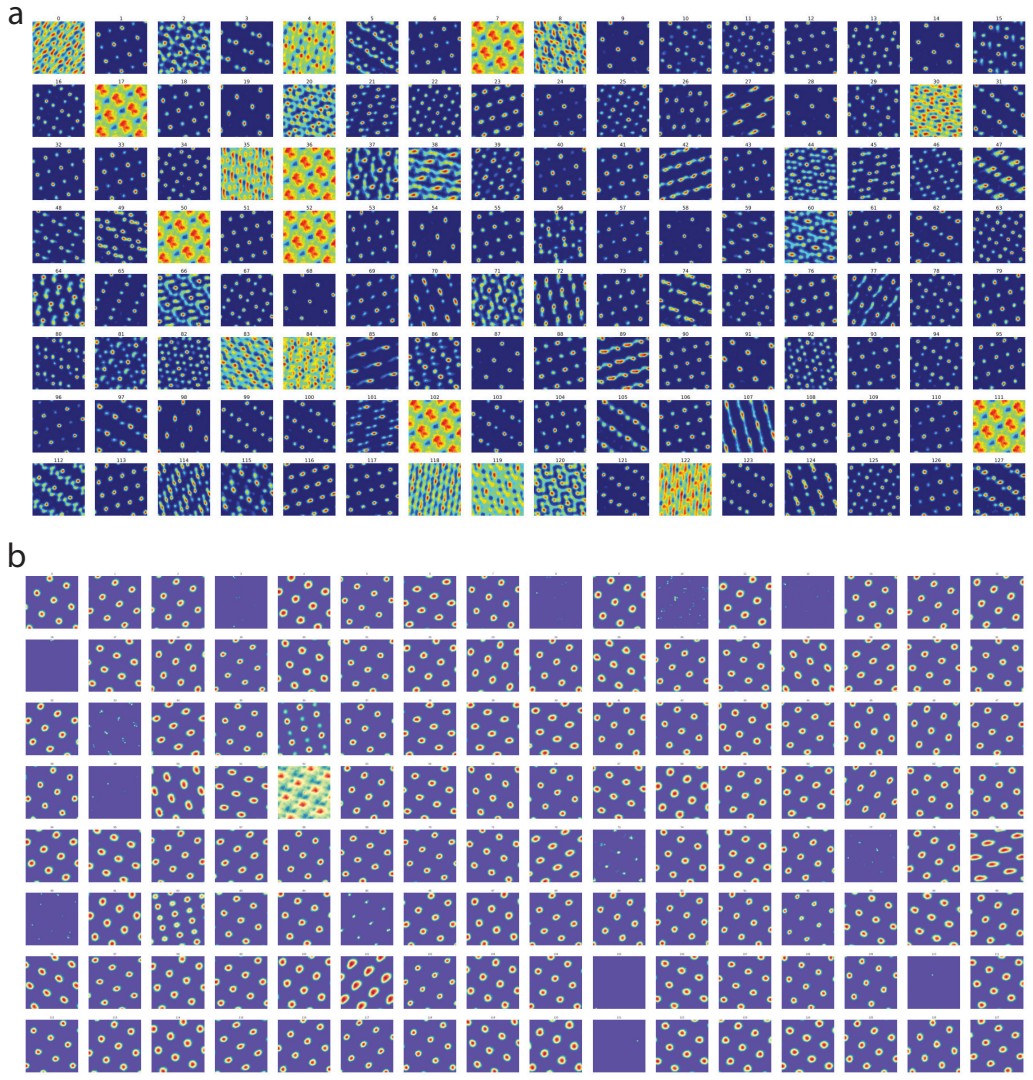

Figure 9: All 128 ratemaps evaluated on trajectories inside a 2m box. (a) Ratemaps from the corresponding to Fig. 4 (b) Ratemaps corresponding to the run in Fig.6

# B All Ratemaps

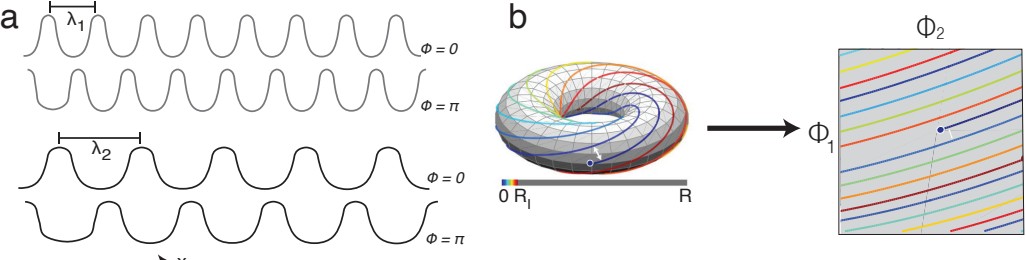

Figure 10: (a) 2 Example cells from 2 modules, with preferred phase $0$ and $\pi$. (b) Visualizing the state space defined by $\{\phi^1, \phi^2\}$ as a torus (left) and on a square with periodic boundary conditions (right), which an equivalent construction of a torus.

## C  Construction of the grid code

To explain the structure of the grid code, we consider idealized tuning curves in 1d.

Each cell $i$ is defined by its periodicity $\lambda^\alpha \in \mathcal{R}$ and preferred phase $\phi_i \in S^1$. All cells in the same module $\alpha$ have the same periodicity and uniformly tile all allowed phases. For position $x \in \mathcal{R}$,

$$r_i^\alpha(x) = R_{\max}\mathrm{ReLU}\left[\cos\left(\frac{2\pi}{\lambda_\alpha}x + \phi_i\right)\right] \tag{9}$$

The tuning curves corresponding to this module can be seen in Fig. 10a.

For this module, we can define $\phi^\alpha(x) = \dfrac{2\pi}{\lambda_\alpha}x$ modulo $2\pi$. Here $\phi^\alpha \in S^1$.

So the firing rate can now be written as

$$r_i^\alpha(x) = R_{\max}\mathrm{ReLU}\left[\cos\left(\phi^\alpha(x) + \phi_i\right)\right] \tag{10}$$

All information about the current state of the module is encoded in the single variable $\phi^\alpha$. Thus the set of phases $\{\phi^\alpha\}_\alpha \in S^1 \times ... \times S^1$ uniquely define the coding states of the set of grid modules.

For 2 modules, defined by $\{\phi^1, \phi^2\}$, these states can be visualized as being on a torus $S^1 \times S^1$, Fig.10b.

## D  Nonlinear Dimensionality Reduction

For Fig. 6, we qualitatively followed the methodology used by seminal experimental papers examining the topology of neural representations [12, 24]: we used principal components analysis to 6 dimensions followed by a non-linear dimensionality (in our case, spectral embedding) reduction to 3 dimensions. Similar results are obtained if one uses Isomap [60].

