# OpenReview forum: "Self-Supervised Learning of Representations for Space Generates Multi-Modular Grid Cells"
_NeurIPS.cc/2023/Conference — NeurIPS 2023 poster_

### Official Review · Reviewer_1Nr3 · 2023-07-03

**Soundness:** 3 good
**Presentation:** 3 good
**Contribution:** 2 fair
**Rating:** 7
**Confidence:** 3

**Summary:**

Shows how grid modules of grid cells can emerge as solutions to a self-supervised learning framework, implemented as a recurrent neural network.
They take insights from Continuous Attractor models (velocity dependent weights), Dorrell et al (2023) representation theory (path invariance), and ideas about efficient coding (Sreenivasan & Fiete), and put them all within a single SSL framework, based on three loss functions for RNNs - maximising separation of distinct locations, path invariance and high capacity for encoding locations.
The simulation results show modules of grid cells are formed.

**Strengths:**

The single theoretical framework for explaining various aspects of the grid code is a strength

**Weaknesses:**

The advance on previous work is not so clear, given that each aspect has been presented previously, with representation theory covered by Dorrell et al., nice analysis of the emergence of grid codes in RNNs in Sorscher et al., and coding efficiency in Sreenivasan and Fiete, and using the basline continuous attractor model of RNNs with velocity dependent weights for path integration.

**Questions:**

Given the eventual aim to understand what is necessary for the emergence of grids in RNNs, should Sorscher et al have been cited earlier? Or the aim to understand the effect of environmental manipulations such as rewards, should Nayebi et al (NeurIPS 2021) be cited?
For the eventual aim of pushing this framework towards machine learning in general domains such as vision, audition etc, what will correspond to velocity, given the reliance on a framework based on path integration?


**Limitations:**

yes

---

> ### Author Rebuttal · Authors · 2023-08-08
>
> We thank Reviewer 1Nr3 for their time and feedback and suggestions. Replying to each comment, suggestion and/or question in turn:
>
> > The advance on previous work is not so clear, given that each aspect has been presented previously, with representation theory covered by Dorrell et al., nice analysis of the emergence of grid codes in RNNs in Sorscher et al., and coding efficiency in Sreenivasan and Fiete, and using the basline continuous attractor model of RNNs with velocity dependent weights for path integration.
>
> We thank the reviewer for the opportunity to better explain why our work is a significant and novel contribution. **_We devoted our entire Global Response to answering this question_**. We refer the reviewer to this response. In short, there are four approaches (that we identify and that you correctly reiterate) and each has at least 1 significant limitation. We use insights from all four perspectives and *solve all their limitations* by contributing a unified model that elegantly combines the strengths of these approaches. Please let us know if you have any follow up questions!
>
> Additionally, the difference between supervised and self-supervised learning is _significant_: In the supervised setting, for each time point, the loss function has access to the absolute position of the agent in the environment. In the self-supervised case, for each pair of positions, the loss function has 1 coarse-grained bit of information: Whether the current position is within $\sigma_x$ of the previously considered position. Consequently, showing a successful solution in self-supervised learning is a significant advance beyond a supervised learning solution.
>
> > Given the eventual aim to understand what is necessary for the emergence of grids in RNNs, should Sorscher et al have been cited earlier?
>
> We’re not quite sure we understand this suggestion. We cite Sorscher et al. 2019 on Page 1 in the second paragraph, after introducing grid cells in the first paragraph, and again on Page 2. Could you please clarify?
>
> > Or the aim to understand the effect of environmental manipulations such as rewards, should Nayebi et al (NeurIPS 2021) be cited?
>
> Nayebi et al. 2021’s primary contribution is regressing candidate artificial neural networks’ activations against mouse electrophysiology recordings from medial entorhinal cortex. As you correctly state, their Section 5 also studies how trajectory statistics (e.g., a preference of moving to a particular location, understood to be a “reward” location) affect the artificial neural population similarly to reward-driven remapping in biological neural populations. Our paper does not regress artificial and biological representations, nor does our paper study how those regressions are affected by trajectory statistics, and so respectfully, we feel Nayebi et al. 2021 is a bit distant to merit a citation.
>
>
> > For the eventual aim of pushing this framework towards machine learning in general domains such as vision, audition etc, what will correspond to velocity, given the reliance on a framework based on path integration?
>
> That’s a great question and one that we see as wide-open! In vision, one might imagine an agent following some “path” in view-space i.e. moving in a way such that the agent returns to the agent’s original vantage point. This can be related to visual saccades produced by primate vision. In audition, one might take inspiration from Aronov, Nevers & Tank Nature 2017, although the analogy is admittedly less clear. This is why we suggest these other modalities as possible future directions.

---

> > ### Author Response · Authors · 2023-08-14
> > **Author Response**
> >
> > Dear Reviewer,
> >
> > Since the author-reviewer discussion period is coming to an end this week, we request the reviewer to consider increasing their score if we have addressed the scientific concerns raised in their review. Specifically, we have devoted our entire global response to better motivate our work and compare it to the previous approaches to grid cell emergence that the reviewer has correctly pointed out. In particular, we clearly state the limitations of each previous approach and show how we have combined insights from all these approaches to overcome every limitation.

---

### Official Review · Reviewer_cRXg · 2023-07-05

**Soundness:** 4 excellent
**Presentation:** 3 good
**Contribution:** 4 excellent
**Rating:** 8
**Confidence:** 3

**Summary:**

The paper proposes a computational framework for the emergence of grid cells in the mammalian cortex through self-supervised learning. The learning objective is formulated combining requirements of path independence for location code, error-correcting coding, efficient coding. Validity of the approach is demonstrated through numerical experiments on a recurrent neural network. Resulting cells reproduce important properties of grid cells observed in biology: cells organized into modules with common spatial frequency and orientation; modules exist for a range of frequencies; cells in a single module regularly tile the space.

**Strengths:**

The paper formulates a principled and biologically meaningful optimization problem and arrives at a representation that manifests important properties of grid cells.

The role of each component of the objective function is studied experimentally.

Rich future directions outlined in Discussion section.


**Weaknesses:**

Authors do not discuss relation of their work to the literature arguing for grid cells role in predictive representation (e.g., Stachenfeld et al, 2017; Momannejad 2020).

The biological plausibility of proposed learning procedure is also not discussed.

**Questions:**

Do all learning batches start from the same implied position x? Otherwise it's hard to imagine coordination of position codes between batches.

Reader needs at least some elaboration on updates of W.

What's a ratemap? Is it a reception field?

I could not decipher figure 5a, left.

line 199-200: do you mean arena is larger in testing than in training? how can the value for the low spatial frequency be found in testing?



**Limitations:**

See weaknesses

---

> ### Author Rebuttal · Authors · 2023-08-08
>
> ## Author Response
>
> We thank Reviewer crXg for their time, positive feedback and high score. Replying to each comment, suggestion and/or question in turn:
>
> > Authors do not discuss relation of their work to the literature arguing for grid cells role in predictive representation (e.g., Stachenfeld et al, 2017; Momannejad 2020).
>
> Stachenfeld et al, 2017; Momannejad 2020 are based on the successor representation (in reinforcement learning) for learning predictive representations, resulting in a place cell-like population code as the successor map) are a normative model for place cell formation. Given a place cell representation, they show that eigendecomposition of their place cell representation results in vectors that are periodic. This point is essentially the same as Dordek et al. 2016 (non-negative PCA of place cells results in grid cells), which was subsequently used by Sorscher et al. 2019 to design better supervised targets for recurrent neural networks. We will add a connection to this line of work with appropriate citations to our Discussion. Thank you for helping us improve the paper!
>
> > The biological plausibility of proposed learning procedure is also not discussed.
>
> To clarify, are you inquiring about the biological plausibility of the data (i.e. the trajectories and their permutations), of the network (i.e. the architecture), of the learning algorithm (i.e. backpropagation), or of something else?
>
> > Reader needs at least some elaboration on updates of W.
>
> Could you please clarify what additional information you’re looking for? W is a square matrix output by an MLP whose inputs are 2D cartesian velocity. We’d be happy to provide whatever additional information once we understand.
>
> Could the reviewer provide us with more specific questions or doubts about W?
>
> > What's a ratemap? Is it a reception field?
>
> Indeed, a ratemap is a receptive field over spatial position. Specifically, one ratemap shows one neuron’s (average) activation values at different spatial locations. A ratemap is computed by partitioning physical space into square bins (typically ~5 cm by 5 cm) and then computing the neuron’s average activation, averaged over all instances that the animal/network passes through the spatial bin.
> Ratemaps are a standard tool in the neuroscience of spatial navigation, used to visualize the spatial tuning of neurons. We will add a small section in the appendix describing the construction of ratemaps.
>
> > Do all learning batches start from the same implied position x? Otherwise it's hard to imagine coordination of position codes between batches.
>
> Yes, all batches start from the same implied position. We have added this clarification in the main text.
>
> > line 199-200: do you mean arena is larger in testing than in training? how can the value for the low spatial frequency be found in testing?
>
> Yes, the testing trajectories are much longer than the training trajectories.

---

> > ### Comment · Reviewer_cRXg · 2023-08-11
> >
> > Thanks for detailed feedback. I have no more issues regarding biological plausibility or W updates.
> > Ratemap: a brief definition of the term in the main text would help the uninitiated reader.
> >
> > On testing: My question is about the size of the arena, which is not the same thing as the trajectory length, right? Larger arena would have areas unexplored during training, and that is my confusion.

---

> > > ### Author Response · Authors · 2023-08-12
> > > **Author response to reviewer cRXg**
> > >
> > > Thank you for your specific question. Generalization to arenas larger than the training arena is a key advance of our work (relative to supervised learning approaches) and is in line with previous theoretical work on grid cell coding.
> > >
> > > The sizes of the arenas in Fig. 5 are larger than the trajectory length, as you have correctly pointed out - this is generalization by extrapolation. Grid modules provide unique positional representations upto a scale that is exponential in the number of modules - see Fiete et al (2008) [1] for theoretical arguments supporting this (also, see below for our intuitive explanation of this paper). Once a multimodular representation has been learnt on the training trajectories, the network can generalize to much larger arenas. The capacity loss provides the key top down inductive bias for this generalization capability (or else the network overfits to the training trajectories, learning a single grid scale which is roughly the size of the training trajectories, as we have shown in our ablation experiments in Fig 7a). This is also why we refer to the capacity loss as "capacity": it allows the network to represent many more locations uniquely.
> > >
> > > In light of these arguments and explanations,if there are no further scientific questions, we would request the review to please consider revising your score.
> > >
> > > We will add a paragraph better contextualizing these results to our paper along with the relevant citation.
> > >
> > > [1] What Grid Cells Convey about Rat Location. Ila Fiete, Yoram Burak and Ted Brookings, Journal of Neuroscience 2 July 2008, 28 (27) 6858-6871
> > >
> > > To intuitively explain this result, multiple modules is akin to decimal or binary: with each additional digit or bit, the number of unique numbers grows exponentially. Grid cells are a little different for two reasons: (1) the scales of each grid module can be of the same order of magnitude, and (2) each digit/bit can be updated in parallel by path integration rather than sequentially (i.e. there’s no need to carryover when updating).
> > >
> > > Another intuitive perspective is a classic number theory argument: the chinese remainder theorem (https://en.wikipedia.org/wiki/Chinese_remainder_theorem):
> > >
> > > For example, to uniquely represent numbers up to 23, one can follow two schemes:
> > >
> > > Scheme 1: Have one box for each number, totalling 23 boxes.
> > >
> > > Scheme 2: Represent the number with remainders after dividing by 3, 5 and 7: This totals 3+5+7=15 boxes. For larger numbers, ratio of boxes between scheme 1 and scheme 2 increases exponentially. For the given example of 3, 5 and 7, one can actually represent numbers up to 3$\times$5$\times$7 = 105 uniquely. You can think of each remainder as one single grid module.

---

### Official Review · Reviewer_dHUQ · 2023-07-05

**Soundness:** 3 good
**Presentation:** 2 fair
**Contribution:** 2 fair
**Rating:** 6
**Confidence:** 4

**Summary:**

This work reviews some of the issues with existing models of grid cells (cells in mammalian brains that fire when the animal is located at the vertices of a hexagonal grid) and suggests a new model based on recurrent neural networks (RNNs). The model is self-supervised, eliminating the worry that the structure of the readout in a supervised task could influence the outcome. Besides leading generically to grid-like firing, the model also exhibits multiple grid scales organized in modules such that the scale is the same within each module but the phase varies.

**Strengths:**

*Originality:*
The paper suggests a way for grid cells to emerge from a self-supervised learning (SSL) paradigm, in contrast to previous work which works mostly in a supervised regime.

*Quality:*
The paper does a good job of surveying prior work and includes a fair amount of simulations.

*Clarity:*
The presentation is generally clear.

*Significance:*
Grid cells are of tremendous interest in neuroscience and there is a considerable volume of work attempting to explain their properties and the reasons behind their existence. This paper provides a novel model for how grid cell may emerge – as a self-supervised means of keeping track of an animal's location in space – and is thus of great interest to neuroscientists.

**Weaknesses:**

1. The authors present this work as a significant advance over methods based on supervised learning because the latter depend on specific design choices. However, the same seems to be true in the new approach: for instance, while the separation and path-invariance loss are pretty natural, the capacity loss is counter-intuitive, as mentioned even by the authors in the Discussion. Excluding the capacity loss eliminates the multi-scale nature of the solution. Moreover, the emergence of grid cells is sensitive to the parameters used in the loss, as shown in Figure 7. It is thus not immediately obvious that the proposed method requires any less fine tuning to lead to grid cells than prior models.
2. Ideally the code used to run all the simulations would have been included with the supplementary material. It was promised for after acceptance, but this seems hard to justify, since the code exists already, and it could be useful for a thorough evaluation of the paper.
3. Most of the figure panels should be significantly larger – Figure 7 is a particularly bad example. I understand that space is a limiting factor, but some progress can be made by including fewer ratemaps (don't see the need for more than 3x3 or 4x2 examples of each kind). Also ensure that font sizes don't go below 7 or 8 – when they do, it may be better to just remove the text because it is very inconvenient to read.

**Questions:**

1. The permutation invariance discussed around eq. (2) assumes Euclidean space, where the order of the steps does not change the outcome of a path. This seems overly restrictive since a typical animal's habitat is unlikely to feature a flat 2d plane, but instead is more likely to contain obstacles, hills, tunnels, etc. Can the authors comment on how their method might adapt to such cases?
2. The motivation for the capacity loss in eq. (9) seems a bit obscure. This was touched upon briefly in the Discussion, but it would be useful for the authors to indicate how they came up with this particular formulation. Also, were other capacity loss functions attempted, and how did they work?

Minor comments:
* line 127: $f$ is used here with a different meaning from $f$ in, e.g., eqns. (2) or (3), which is a bit confusing
* Figure 6, panel c: what non-linear dimensionality reduction technique was used?

**Limitations:**

The authors have adequately discussed limitations of their work.

---

> ### Author Rebuttal · Authors · 2023-08-08
>
> ## Author Response
> We thank Reviewer dHUQ for their time and feedback. Replying to each comment, suggestion and/or question in turn:
>
> > The authors present this work as a significant advance over methods based on supervised learning because the latter depend on specific design choices. However, the same seems to be true in the new approach [...] It is thus not immediately obvious that the proposed method requires any less fine tuning to lead to grid cells than prior models.
>
> Thank you for giving us a chance to clarify! We want to make three points.
> 1. Firstly, the prior supervised path-integrating network papers are insightful and valuable. Our paper would not be possible without their contributions.
> 2. Secondly, we want to reiterate that our paper is a significant contribution not just to supervised path-integrating networks but to three other lineages of understanding grid cells: basis function optimization, coding theory and continuous attractor models . As we explain in our Global Response, all four lines of work have at least one significant limitation, and our paper solves all four lineages’ limitations simultaneously. This is why we feel our paper is a novel & significant contribution.
> 3. Thirdly, the limitations of supervised path-integrating networks that we identify are more nuanced than they “depend on specific design choices.” To explain generally the shortcomings we see with supervised path-integrating networks, there are several. See Approach 2 of the Global Response for an extensive discussion.
>
> > the capacity loss is counter-intuitive, as mentioned even by the authors in the Discussion.
> > The motivation for the capacity loss in eq. (9) seems a bit obscure. This was touched upon briefly in the Discussion, but it would be useful for the authors to indicate how they came up with this particular formulation.
>
> We thank the reviewer for the opportunity to better explain the capacity loss. The capacity loss is one of our conceptual breakthroughs, and a key component of our paper. Previous theory work [Fiete, Burak, Brookings 2008; Sreenivasan & Fiete 2011] identified a high coding capacity as one of the key properties the grid code provides.
>
> The capacity loss demands from the network: “Use as little of your total available coding volume as possible on the training data (subject to separation and path invariance).” This is linked to generalization. On the training data, the network learns how to dynamically evolve its representations in a manner consistent with velocity inputs. The network could learn a ‘shortcut solution’ (single scale solution) and use up all its coding volume on the training data, but then it will fail to generalize to longer trajectories (Fig. 3eg). The capacity loss provides a top-down inductive bias to prevent this ‘shortcut solution’.
>
> > Also, were other capacity loss functions attempted, and how did they work?
>
> No, no other capacity loss functions were attempted. We can explain how this capacity loss came to be. We found that the first two loss terms reliably produced one grid module (i.e. all units sharing one period, as shown in our ablation figure) but that the period scaled to match the training trajectory length. The problem is that there’s no incentive to ensure that a location outside the training distribution has a different representation than a location inside the training distribution, but that’s precisely what we needed. How could we incentivize generalization? Fig 3efgh was the conceptual breakthrough: Fig3e and Fig3f both achieve the same _training_ loss, but only Fig3f generalizes. Thus, we needed a loss term to prefer Fig3f over Fig3e, and once we conceptualized Fig 3e and 3f the first loss function we thought of was the paper’s current capacity loss.
>
> > 2. Ideally the code used to run all the simulations would have been included with the supplementary material. It was promised for after acceptance, but this seems hard to justify, since the code exists already, and it could be useful for a thorough evaluation of the paper.
>
> While NeurIPS does encourage submitting code, doing so is not a requirement and it seems unfair to penalize us for failing to do something that is not required. Moreover, submitting code is not without risk: unintentionally failing to completely anonymize the code can result in an immediate rejection.
>
> > 3. Most of the figure panels should be significantly larger – Figure 7 is a particularly bad example.
>
> Agreed. We will improve the figures as suggested.
>
> > 1. The permutation invariance discussed around eq. (2) assumes Euclidean space [...] This seems overly restrictive since a typical animal's habitat is unlikely to feature a flat 2d plane [...] Can the authors comment on how their method might adapt to such cases?
>
> We thank you for raising this question! To the best of our knowledge, previous modeling work typically assumes flat 2d Euclidean space.  It is certainly an interesting point of discussion. Grid cells have been observed in bats navigating in 3D (Yartsev & Ulanovsky 2011 and Ginosar et al. 2021) and at least one paper has studied grid cells in higher dimensional spaces (Klukas, Lewis, Fiete 2020). However, in this work, we follow the majority of previous modeling work that focuses on flat 2D Euclidean space. We leave 3D for future work. We have added this to our discussion section.
>
> > Figure 6, panel c: what non-linear dimensionality reduction technique was used?
>
> We qualitatively followed the methodology used by seminal experimental papers examining the topology of neural representations e.g., Chaudhuri et al. Nature neuroscience 2019 and Gardner et al. Nature 2022: we used principal components analysis to 6 dimensions followed by a non-linear dimensionality reduction to 3 dimensions (in our case, spectral embedding). Similar results are obtained if one uses Isomap.  We will add an explanation and citations to these works in the appropriate section.

---

> > ### Comment · Reviewer_dHUQ · 2023-08-11
> >
> > Thanks for the explanation!
> >
> > Regarding sharing the code, a couple of points: first, I don't think my score would have been different had you included it. That does not mean the lack of the code is not a weakness. Second, had NeurIPS *required* the code to be submitted, the paper wouldn't have just been penalized for not including it, but it would have been rejected. That of course is not the case. It seems reasonable to me to be penalized for failing to do something you "are strongly encouraged" to do (to quote from the submission guidelines)—indeed, it would seem like a rather empty suggestion if it had no incentive to push authors to follow it. Finally, I sincerely doubt any work would be rejected for unintentionally sharing your identities in shared code. Most of the submissions I've reviewed have had code included and everything went fine.

---

> > > ### Author Response · Authors · 2023-08-12
> > > **Author response to reviewer dHUQ**
> > >
> > > Thank you. For future submissions, we will ensure we have code with the submission.
> > >
> > > Here, we would like to take the opportunity to reiterate the difference between a supervised and self-supervised approach: In the supervised learning approaches (e.g. Banino et al, 2018; Sorscher et al, 2019), for each time point, the loss function has access to the absolute position of the agent in the environment. It’s unrealistic to assume that biological agents have access to their absolute spatial position at all times. In the self-supervised case, for each pair of positions, the loss function has 1 coarse-grained bit of information: Whether the current position is within $\sigma_x$ of the previously considered position. This is the key difference between the 2 approaches and the fact that our self-supervised networks learn multiple modules of grids despite having access to only this impoverished version of spatial information is the significant, non-trivial advance. We have varied hyperparameters of the loss (specifically the coding scale $\sigma_g$) to show the different representations that can emerge - unlike supervised approaches that claim generality far beyond their empirical results.
> > >
> > > If there are no further scientific questions, please consider revising your score.

---

> > > > ### Comment · Reviewer_dHUQ · 2023-08-13
> > > >
> > > > Thank you for explaining the advantages of the self-supervised approach again. I will update the score from 5 to 6.

---

> > > > > ### Author Response · Authors · 2023-08-18
> > > > > **Author response**
> > > > >
> > > > > We appreciate the discussion and engagement with our work and the increase of your score from 5 to 6.
> > > > >
> > > > > We'd like to emphasize two central aspects of our work's novelty. We kindly ask you to further increase your score if these insights and perspectives resonate with you:
> > > > >
> > > > > 1) Capacity: One of the primary contributions of the work lies in the introduction of capacity as a loss term in our self-supervised learning approach. We believe that capacity is an essential factor to consider, especially when aiming to model biological neural representations where efficient representations are crucial. Capacity was considered only by grid cell theorists and by explicitly constructing our capacity loss term, we have demonstrated how to incorporate it as a broadly applicable normative principle for neuroAI. To the best of our knowledge, this is a novel concept that hasn’t been explored in previous works. Furthermore, the potential applicability of this concept extends beyond just our current work; it might be highly relevant to other modalities within machine learning and neural modeling where efficiency and capacity are of paramount importance.
> > > > >
> > > > > 2) Self-Supervised Learning for Spatial Navigation: Another significant aspect of our study is the application of self-supervised learning to spatial navigation. Spatial navigation, as a cognitive task, has its roots deeply intertwined with abstract reasoning in the brain. By applying self-supervised learning techniques to this domain, we hope to bridge the gap between embodied computational models and biological systems, potentially offering insights into how animals, including humans, make consistent models of their environment. We believe that our work is the first to offer a self-supervised approach in this domain, thereby contributing a unique perspective to this problem.
> > > > >
> > > > > We trust that our contributions, when viewed in light of the above perspective, underscore the novelty and potential impact of our research.

---

> > > > > > ### Comment · Reviewer_dHUQ · 2023-08-18
> > > > > >
> > > > > > Thanks for the additional explanations. I will keep the score as-is.

---

> > > > > > > ### Author Response · Authors · 2023-08-19
> > > > > > > **We have now provided anonymized code**
> > > > > > >
> > > > > > > Thank you for your reply.
> > > > > > >
> > > > > > > We have now anonymized our code and made it available at this anonymous google drive link:
> > > > > > >
> > > > > > > https://drive.google.com/drive/folders/1JNmdeTpJhktOoFJ-slC1l2AIqRSw3sAk?usp=drive_link
> > > > > > >
> > > > > > > Let us know if this code helps in your assessment of the robustness of our results.
> > > > > > >
> > > > > > > Thank you,
> > > > > > >
> > > > > > > Authors.

---

### Official Review · Reviewer_keKQ · 2023-07-05

**Soundness:** 3 good
**Presentation:** 4 excellent
**Contribution:** 3 good
**Rating:** 7
**Confidence:** 3

**Summary:**

The paper shows that recurrent networks trained with a "self-supervised" loss leads to units of the internal representations that organize as grid cells. In particular, the paper defines a loss that promotes separations between neural representations encoding different spatial locations, encourages a representation to be invariant to different possible paths taken leading to it's representation, and maximizing the capacity of the representation, and the authors use paired velocities and neural representations as their dataset.


**Strengths:**

- The paper was very clearly written and organized, with nice visuals that supported the text. Moreover, the paper provided a helpful background of previous research that was relevant to the formulation used in the paper.
- The paper defines a loss that is nicely linked to existing theories for the emergence of grid cells, and shows that this loss, optimized using gradient descent leads emergence of grid cells in artificial recurrent networks.
- The authors performed ablations of the hyperparameters in their loss to show the dependence of their results on the different terms

**Weaknesses:**

- How robust are the results to other hyperparameters, like batch-size, learning rate, etc?
- While the authors criticize previous work that identified the emergence of grid cells using supervised RNN to specifics of the target function (line 55), the authors do not seem to properly explain how their setup differs, and does not lead to similar implicit assumptions. For example, what is the difference between the velocities being used as a supervised signal (which the authors criticize), versus incoporating implicitly into their dataset and self-supervised loss? Are there similar assumptions with respect to the creation of the dataset (e.g having a sufficient number of examples with overlapping positions)
- Further, the authors assert that "SSL mitigates the need for large scale supervised data", but it is unclear to me how different it is to incorporate the velocities as a paired dataset rather than a target variable for a supervised objective.
- (Minor) The claim in the discussion "how might ... SSL principles be applied to drive computational neuroscience forward" seems too general.
- (Minor) Difficult to read text in Figure 7
- (Minor) extra italics on t in line 138

**Questions:**

- Do all the internal units have grid cell properties, or only a fraction?
- Related, but rather than show example units that look like grid cells, are there any metrics that quantify the extent to which a cell is a grid cell?
- Are there any predictions that can be made, for example, what might happen for animals that explore 3d space?
- Can the authors comment on the relationship between the toroid's and the encoding of space to make the paper self-contained (e.g. schematics from Fig. 2)?

**Limitations:**

Yes

---

> ### Author Rebuttal · Authors · 2023-08-08
>
> ## Author Response
>
> We thank Reviewer keKQ for their time, feedback and high score.
>
> Replying to each comment, suggestion and/or question in turn:
>
> > How robust are the results to other hyperparameters, like batch-size, learning rate, etc?
> > e.g having a sufficient number of examples with overlapping positions
>
>
> This is a good question. We have some initial answers but more work is necessary and already underway! For context, SSL is notoriously compute hungry (see Balestriero et al. 2023), and training one of our SSL networks requires a 16 GB GPU for ~7 days, whereas a supervised path-integrating network from previous papers takes an ~8 GB GPU for ~6 hours.
>
> We have found that batch sizes of 120, 130 and 150 all work equally well; we haven’t tried smaller batch sizes, and larger batch sizes can result in OOM errors. We tested learning rates in {0.002, 0.0002, 0.00002} x optimizers in {Adam, AdamW, NAdam, NAdamW} x learning rate schedulers in {Reduce LR on Plateau, Linear Warmup with Cosine Annealing} and found different combinations have different learning curves but reach the same result. We’re currently running additional experiments and ablations.
>
> Regarding sufficiently many overlapping positions, we have not yet tested this rigorously. We conjecture that some fraction of overlapping positions is necessary for two reasons: (1) no overlapping positions is akin to contrastive SSL without a sufficient number of positive pairs, and (2) in the past, we tried sampling trajectories without overlapping positions and random representations emerged because the network does not need to path integrate; rather, it only needs to ensure it never repeats a code word. We intend to test what fraction of overlapping points is necessary to put in the appendix but have not yet been able to prioritize that experiment.
>
>
> > While the authors criticize previous work that identified the emergence of grid cells using supervised RNN to specifics of the target function (line 55),
>
> Line 55 was unintentionally harsh. We will rewrite this.
>
> >  the authors do not seem to properly explain how their setup differs, and does not lead to similar implicit assumptions.
>
> This is good feedback! We will do a significantly better job at distinguishing our approach from the previous supervised learning approach. We have devoted our Global Response (Approach 2) to this point.
>
>  - To your question about distinguishing ourselves from previous approaches, please see the Global Response.
>
>  - To explain generally the shortcomings we see with supervised path-integrating networks see Approach 2 of the Global Response
>
> > Further, the authors assert that "SSL mitigates the need for large scale supervised data"
>
> To contextualize that sentence, the sentence is in our Background and reads: “SSL is increasingly gaining popularity as a normative approach in neuroscience since SSL mitigates the need for large-scaled supervised data that biological agents do not possess.” We think this is a general viewpoint across multiple modalities (vision, language, audition) that many neuroscience & ML papers have made previously. We will add citations to prominent neuroAI work that uses self-supervised learning across modalities that make a similar point.
>
> In the particular modality of spatial navigation, we can comment on the feasibility of the self-supervised versus supervised setup. Recall that the goal is to learn a self-consistent representation of spatial position that is updated by velocity. It’s unrealistic to assume that biological agents have access to their absolute spatial position at all times; if they did, why would they need to learn their spatial position? Consequently, the supervised target bypasses the learning problem. In contrast, permuting trajectories is generally straightforward. For instance, if you walk to a grocery store twice, you might go North then West on the first trip, and you might go West then North on the second trip. We think our setup is much more realistic for biological agents.
> To further contrast supervised & SSL, we do not include velocities as a paired dataset. Rather we use velocities and convert it to a binary tensor of 1’s and 0’s, where 1 denotes that positions are within $\sigma_x$ of each other. This is a very low information, coarse-grained and impoverished learning target.
>
> >  are there any metrics that quantify the extent to which a cell is a grid cell?
>
> There is indeed a metric! Unfortunately, the “grid score” used by previous works (such as Banino et al. 2018, Nayebi et al. 2021, Schaeffer et al. 2022) are applicable either for _hexagonal_ lattices or _square_ lattices but our multi-periodic lattices are all sheared. This means that the square or hexagonal grid score cannot be used for our networks to identify grid-like units.  We suspect that the ``conformal isometry” loss function [first identified by Dehong Xu, Ruiqi Gao et al. 2022] might make our lattices more hexagonal and we are currently exploring this direction.
>
>
> > Predictions for 3d space
>
> We thank you for this question! To the best of our knowledge, the majority of previous work for grid cells assumes flat 2d Euclidean space.  It is certainly an interesting point of discussion. Grid cells have been observed in 3D [Yartsev 2011, Ginosar et al. 2021] and at least one paper has studied them in higher dimensional spaces [Klukas et al. 2020]. However, we follow the majority of previous modeling work that focuses on 2D Euclidean space. We leave 3D for future work; we will add this to our Discussion.
>
> >  The claim in the discussion "how might ... SSL principles be applied to drive computational neuroscience forward" seems too general.
>
> We’ll rework this sentence. We weren’t trying to say something grandiose, merely that we hope our paper might be useful to the field.
>
> > Difficult to read text in Figure 7
>
> We will make the fonts in all our figures larger.

---

> > ### Author Response · Authors · 2023-08-14
> > **Author response**
> >
> > Dear Reviewer,
> >
> > Since the author-reviewer discussion period is coming to an end this week, we request the reviewer to consider increasing their score if we have addressed the scientific concerns raised in their review. Specifically, we have expanded on the difference between a supervised and self-supervised approach in the Global Response.
> >
> > Here, we reiterate the key difference: In the supervised learning approaches (e.g. Banino et al, 2018; Sorscher et al, 2019), for each time point, the loss function has access to the absolute position of the agent in the environment. It’s unrealistic to assume that biological agents have access to their absolute spatial position at all times. In the self-supervised case, for each pair of positions, the loss function has 1 coarse-grained bit of information: Whether the current position is within $\sigma_x$ of the previously considered position. This is the key difference between the 2 approaches and the fact that our self-supervised networks learn multiple modules of grids despite having access to only this impoverished version of spatial information is the significant, non-trivial advance.

---

> > ### Comment · Reviewer_keKQ · 2023-08-18
> >
> > I thank the authors for their reply. I remain concerned regarding the robustness of the results, which seems to arise from the lack of quantitative metric characterizing their grid cells; and instead only providing some example units from the different runs and ablations (without quantifying the fraction of cells with "grid-like" properties). I also suspect this may make characterizing the effect of the machine learning hyperparameters more difficult as well, once those runs complete. Given the lack of quantitative demonstration of the robustness of the results, it makes it difficult to gauge the significance of the findings.

---

> > > ### Author Response · Authors · 2023-08-19
> > > **We have now provided anonymized code**
> > >
> > > Thank you for your reply.
> > >
> > > We have now anonymized our code and made it available at this anonymous google drive link:
> > >
> > > https://drive.google.com/drive/folders/1JNmdeTpJhktOoFJ-slC1l2AIqRSw3sAk?usp=drive_link
> > >
> > > Let us know if this code helps in your assessment of the robustness of our results.
> > >
> > > Thank you,
> > >
> > > Authors.

---

### Author Rebuttal · Authors · 2023-08-08

## Global Response to Reviewers

Here we take the opportunity to better motivate our paper, and to explain why we view it as a novel and significant contribution. In short, multiple approaches have been taken to understand grid cells, but each contains at least one limitation. Our paper combines their strengths to simultaneously solve their limitations.


Broadly, there are four approaches to understanding grid cells:


**Approach 1: Basis Function Optimization** (e.g. Dorrell et al. 2023)

**Perspective:** Given functions of space constructed by linear combinations of sine and cosine basis functions, provides loss function(s) that makes these functions look grid-like.

**Limitation(s):**

1. The central question is how one should learn a self-consistent representation of space updated by velocity inputs. This learning of an attractor with a continuum of fixed points is known to be a hard problem, covered in both classic papers (e.g. Seung 1996) and recent reviews (e.g. Khona and Fiete 2022). By explicitly starting with functions of space, Dorrell et al. 2023 avoid this problem.
2. There is no neural network component - this approach is pure basis function optimization
3. The assumed sine and cosine basis functions lie in the correct function class to learn periodic representations and thus provide an advantageous inductive bias that artificial networks lack.


**Approach 2: Supervised Reconstruction of Supervised Spatial Targets by recurrent networks** (e.g., Banino et al 2018, Sorscher et al. 2019)

**Perspective:** Supervised learning on spatial target functions sometimes leads to grid-like representations.

**Limitation(s):**

1. By assuming access to supervised learning targets encoding privileged absolute spatial position, these approaches again bypass the central question. If biological agents already had their own absolute spatial position, they wouldn’t need to learn how to track their own absolute spatial position.
2. The design choices between our work and previous works are qualitatively different. Previous supervised learning papers crafted supervised targets to insert grid-like representations into the networks, sometimes contradicting known biological properties of place cells (Schaeffer et al. NeurIPS 2022). In contrast, our learning setup is motivated by first-principles properties of the grid representation identified by previous theory work.
3. Previous supervised papers claimed extreme generality, far beyond what the numerical results support. Schaeffer et al. 2022 showed that their results are tuned and their theory only sometimes predicts empirical results. The criticism of supervised path-integrating networks is this mismatch between claims and empirical results. We were very careful to write our paper in the language of “There exists” and not in the language of “For all” and have avoided using sweeping terms such as “generically” to describe our results.
4. Lastly, our results are different. Despite best efforts, previous supervised path-integrating recurrent neural networks did not learn multiple grid modules. In contrast, our networks do learn multiple modules. Additionally, our networks generalize outside the training distribution, whereas supervised networks fare poorly away from the training arena box where supervised targets have not been defined.

Thus, this line of work is not about the task of path integration, but rather about the shape of targets to obtain grid-like tuning curves. Stachenfeld et al. 2017 and Momennejad et al. 2020 are related in that given a place cell representation (derived from a predictive approach), they show that eigendecomposition of their place cell representation results in vectors that are multiperiodic. This point is essentially the same as Dordek et al. 2016 (non-negative PCA of place cells results in vectors that are grid-like), which was subsequently used by Sorscher et al. 2019 to design better supervised targets for recurrent neural networks and was analytically shown.

**Approach 3: Coding Theory** (e.g., Srinivasan and Fiete 2011)

**Perspective:** Given the grid cell representation, what are the unique coding theoretic properties that this representation provides?

**Limitation(s):**

Lists many of the coding-theoretic properties of grid cells, but doesn’t test the necessity or sufficiency of those properties for generating grid cells via an optimization problem or provide explicit loss function(s).


**Approach 4: Continuous Attractors** (e.g., Burak and Fiete, 2009)

**Perspective:** What is the physical principle underlying grid cells? How do we build a mechanistic model that reproduces grid-like patterns using this principle?

**Limitation(s):**

1. Does not provide an optimization problem that leads to grid cells, rather builds a mechanistic model by hand. Neuroscientists would say that there is no “normative” answer to why mammals have learned this particular representation over an evolutionary time scale.
2. Produces only a single module of grid cells.
3. This approach relies on the principle of pattern formation and continuous attractor dynamics: This needs the interaction kernel between neurons to be fine-tuned to produce a continuum of fixed points.


Our work uses insights from all four approaches and solves their limitations by contributing a unified model that elegantly combines their strengths. *We provide an optimization problem motivated from a normative perspective that does not require privileged access to supervised position information, such that networks trained on the optimization problem reliably learn multi-modular periodic representations and generalize significantly beyond their training distribution.*

---

### Decision · Program_Chairs · 2023-09-21

**Decision:**

Accept (poster)

**Comment:**

The paper shows that a self-supervised learning objective can generate a multi-modular grid cell code for representing space. All reviewers agree that the paper tackles an important question and makes a valuable contribution. One of the main concerns raised during the review and discussion period was how robust this finding is. While this issue was not fully resolved by experiments, the paper still appears to be a valuable contribution and code will be available for others to build upon the work.